Continuum beliefs in the stigma process regarding persons with schizophrenia and depression: results of path analyses

Makowski Anna C. 1 a.makowski@uke.de
Mnich Eva E. 1
Angermeyer Matthias C. 2 3
von dem Knesebeck Olaf 1
1 Department of Medical Sociology, University Medical Center Hamburg-Eppendorf , Hamburg , Germany
2 Center for Public Mental Health , Gösing am Wagram , Austria
3 Department of Clinical and Molecular Medicine and Public Health, University of Cagliari , Cagliari , Italy
Morris Richard
Electronic publication date: 2016 Sep 27
Publication date: 2016
Volume: 4
Electronic Location ID: e2360
Received 2016 May 5; Accepted 2016 Jul 22
Copyright: © 2016 Makowski et al.
Copyright year: 2016
Copyright holder: Makowski et al.
License: This is an open access article distributed under the terms of the Creative Commons Attribution License, which permits unrestricted use, distribution, reproduction and adaptation in any medium and for any purpose provided that it is properly attributed. For attribution, the original author(s), title, publication source (PeerJ) and either DOI or URL of the article must be cited.
License URL: https://creativecommons.org/licenses/by/4.0/

Keywords: Continuum belief, Public attitudes, Stigma process, Depression, Schizophrenia

Funding: German Federal Ministry of Education and Research 01KQ1002B Psychenet is a project network funded by the German Federal Ministry of Education and Research (funding code 01KQ1002B). The funders had no role in study design, data collection and analysis, decision to publish, or preparation of the manuscript.

==============================
Background

Individuals with mental illness often experience stigmatization and encounter stereotypes such as being dangerous or unpredictable. To further improve measures against psychiatric stigma, it is of importance to understand its components. In this study, we attend to the step of separation between “us” and “them” in the stigma process as conceptualized by Link and Phelan. In using the belief in continuity of mental illness symptoms as a proxy for separation, we explore its associations with stereotypes, emotional responses and desire for social distance in the stigma process.

Methods

Analyses are based on a representative survey in Germany. Vignettes with symptoms suggestive of schizophrenia (n = 1,338) or depression (n = 1,316) were presented to the respondents, followed by questions on continuum belief, stereotypes, emotional reactions and desire for social distance. To examine the relationship between these items, path models were computed.

Results

Respondents who endorsed the continuum belief tended to show greater prosocial reactions (schizophrenia: 0.07; p < 0.001, depression: 0.09; p < 0.001) and less desire for social distance (schizophrenia: −0.13; p < 0.001, depression: −0.14; p < 0.001) toward a person with mental illness. In both cases, agreement with the stereotypes of unpredictability and dangerousness was positively associated with feelings of anger and fear as well as desire for social distance. There were no statistically significant relations between stereotypes and continuum beliefs.

Discussion

Assumptions regarding continuum beliefs in the stigma process were only partially confirmed. However, there were associations of continuum beliefs with less stigmatizing attitudes toward persons affected by either schizophrenia or depression. Including information on continuity of symptoms, and thus oppose perceived separation, could prove helpful in future anti-stigma campaigns.

Introduction

Despite a progress in evidence-based treatments and increasing public knowledge about mental disorders, persons with mental illness still report direct experience of stigma and discrimination (Alonso et al., 2009). Internalized stigma is negatively associated with lower self-esteem, empowerment and treatment adherence (Livingston & Boyd, 2010) and individuals with mental health problems still encounter stereotypes. Negative emotional reactions as well as desire for social distance toward persons affected have even increased over the last decades (Angermeyer et al., 2014a). Previous research in Germany on the public stereotype of schizophrenia has found the ascription of being unpredictable the most prevalent, followed by the perception of dangerousness, while blaming the person for the illness was the least common. While dangerousness and being unpredictable were strong predictors of desire for social distance, blaming the person was associated with the acceptance of structural discrimination (Angermeyer & Matschinger, 2004). Similar results on the prevalence of stereotypes have also been obtained by a more recent study evaluating the effects of a documentary film on stigmatization (Thonon et al., 2016). The concern about social rejection and insufficient mental health literacy has led to various public campaigns such as “beyond blue” in Australia (Jorm, Christensen & Griffiths, 2005), “Time to Change” in Great Britain (Evans-Lacko, Henderson & Thornicroft, 2013), the German campaigns “Nuremberg Alliance against Depression” (Hegerl, Althaus & Stefanek, 2003) and “psychenet–Hamburg Network for Mental Health” (Makowski et al., 2016) or “Schizophrenia has many faces” in Austria (Grausgruber et al., 2009). The evaluation of the different campaigns came to inconsistent results due to different outcome parameters and indicators used. Some reported improvement in intended behavior but not for knowledge and attitudes (Evans-Lacko, Henderson & Thornicroft, 2013), others showed only minor changes in attitudes (Makowski et al., 2016) or even found an increase in desire for social distance over the course of time (Grausgruber et al., 2009).

In order to further improve strategies to reduce stigma, it is of importance to understand its components and associated factors. Link & Phelan (2001) have conceptualized the stigma process model consisting of several distinct, but interrelated steps. In this conceptualization, stigma exists when the following components converge: 1) distinguishing and labeling human differences, which is a natural selection as soon as it comes to differences that matter socially; 2) Dominant cultural beliefs link these labeled differences to undesirable characteristics (stereotypes); 3) Labeled persons are placed in distinct categories to separate “us” from “them”; 4) Labeled persons experience status loss and discrimination. Additionally, Link et al. (2004) incorporated emotional responses into this process, which are critical to understanding the behavior of stigmatizer and recipients of stigmatizing reactions. Labeling, stereotyping and separation are likely to be associated with emotions such as anger, anxiety and pity. These emotional responses may shape subsequent behavior toward the stigmatized person. This sequence of emotional response and performed conduct has also been postulated by Weiner (1985) in attribution theory.

A lot of research has attended to the stigma process. However, the focus was mostly on single components or the interaction of a few steps in the stigma process. Link et al. (2004) have reviewed empirical articles published on the stigma of mental illness between 1995 and 2003 and took note which concepts had been covered. Most articles were published on stereotyping, followed by status loss/discrimination, emotional reactions and cognitive separation (explained by the authors as fundamental differences between “us” and “them” implied by social labels). In a study by Angermeyer & Matschinger (2005), labeling a person in a vignette as mentally ill (in this case schizophrenia) was associated with an increased likelihood to be tied to stereotypes such as dangerous or unpredictable, which led to greater desire for social distance. Moreover, the authors found that labeling someone as mentally ill and perceiving this person as dangerous was closely associated with feelings of fear and anger (Angermeyer & Matschinger, 2003). Martin, Pescosolido & Tuch (2000) also showed that respondents who labeled a vignette as mental illness expressed greater desire to socially distance themselves. For the case of schizophrenia, a review by Read et al. (2006) found that diagnostic labeling by the public was associated with greater desire for social distance as well as fearful reactions. This is also corroborated by Jorm & Griffiths (2008), who found associations between a biomedical conceptualization of mental illness and a belief in dangerousness for schizophrenia. Another central step in the stigma process, the separation between “us” and “them” has only scarcely been researched. Only few studies were found which explicitly dealt with a continuum concept of mental (ill-) health. This concept implicates that mental health and illness can be located on a continuum rather than a clear, dichotomized separation. Regarding the public’s agreement or disagreement to continuity of mental illness, Schomerus, Matschinger & Angermeyer (2013) found that belief in a continuum was related to decreased feelings of fear, but greater feelings of anger. In all disorders under study (depression, alcohol dependence, schizophrenia), continuum beliefs were significantly associated with greater pro-social reactions and lower desire for social distance. Similar results were published by Angermeyer et al. (2014b), who also found associations between continuum beliefs and positive emotional reactions as well as decreased desire for social distance in the case of depression and schizophrenia. Wiesjahn et al. (2014) examined the association between continuum belief and stigmatization using the Continuum Beliefs Questionnaire. Higher levels of continuum beliefs were associated with lower levels of stereotypes. However, there were no associations with desire for social distance.

Against this background, we set out to explore the relationship of four consecutive components of the stigma process, with a particular focus on continuum beliefs. The aim of this study is to incorporate the belief in continuity of mental illness symptoms into the stigma process model (Link & Phelan, 2001; Link et al., 2004) and to estimate the associations using a path modeling approach. We assume the belief in continuity of symptoms to be a proxy for the step of separation in the stigma process.

Figure 1 displays the theoretical considerations based on the stigma process model. We assume the following: Positive association between stereotype and desire for social distance.

Negative association between stereotype and continuum belief.

Positive association between stereotype and feelings of anger and fear; negative association between stereotype and positive emotional reactions.

Negative association between continuum belief and desire for social distance.

Positive association between continuum belief and positive emotional reactions, negative association between continuum belief and feelings of anger and fear.

Negative association between positive emotional reaction and desire for social distance, positive association between feelings of anger and fear and desire for social distance.

Figure 1 Theoretical model based on the stigma process as postulated by Link & Phelan (2001) (Link et al., 2004).

Methods

Study design and sample

Analyses were based on a telephone survey (CATI–computer assisted telephone interview), which had been conducted in the German cities Hamburg and Munich in the spring of 2014. The survey was part of psychenet–Hamburg network for mental health, a joint project in the metropolitan area of Hamburg. Major component of psychenet was an information and awareness campaign on mental health, including mental disorders (Härter et al., 2012). One purpose of the survey was to evaluate possible effects of the campaign in Hamburg, Munich served as control region.

The sample was comprised of adults aged 18 and older, living in private households in one of the two cities with access to conventional telephone connections. Their numbers were drawn from all registered telephone numbers at random; ex-directory households were also included via computer-generated numbers. The study was approved by the Ethics Commission of the Medical Association in Hamburg (PV3707). In total, 2,006 respondents agreed in participating; this reflects a response rate of 53%. Informed consent was considered to have been given when the individuals completed the interview. Comparisons with official statistics showed that the distribution of demographic characteristics such as gender, age and education is similar to that in the general population in Hamburg and Munich (Mnich et al., 2015). In our analyses, we focus on two subsamples which had been presented either a schizophrenia (n = 1,338) or depression vignette (n = 1,316). While the case stories contained signs and symptoms suggestive of the disorders, they did not include diagnostic labels (please see annex for vignettes). The vignettes had been developed with the input of clinicians based on ICD-10 and DSM-IV criteria and audio-recorded with a trained speaker to increase reliability and to counteract possible interviewer effects. Gender of the “patient” was systematically varied. The vignette was presented in the beginning of the interview, directly followed by the question on continuum belief. Then, the interviewer asked the respondents to name the disorder in the vignette. If the interviewees were not able to identify the disorder correctly, they were provided with the response. Subsequently, they were asked questions regarding mental health literacy (e.g. prevalence of the disorder, possible treatment options) and attitudes, which are described in detail in the next section. Sociodemographic characteristics of the subsamples are described in Table 1.

Table 1 Sociodemographic characteristics of the subsamples in %.

	Subsample schizophrenia (n = 1,338)	Subsample depression (n = 1,316)	
Sex (Female)	52.3	51.8	
Level of education	
 up to 9 years	33.4	31.3	
 10 years	23.0	23.9	
 12–13 years	43.7	45.0	
Age groups	
 18–25	10.1	12.4	
 26–45	39.9	39.2	
 46–65	30.0	28.2	
 > 65	20.0	20.2	
Mean age (SD)	47.7 (18.1)	47.2 (18.3)	

Instruments

Continuum belief

Immediately following the presentation of the vignette, the interviewer posed a question on the respondents’ belief in a continuum of symptom experience. They were asked to indicate their (dis-)agreement to the following statement: “Basically, we are all sometimes like this person. It is just a question how pronounced this state is.” Answers were given on a 4-point Likert-scale ranging from 1 “completely disagree” to 4 “completely agree” plus “don’t know” category.

Stereotypes

Based on previous research (Angermeyer & Matschinger, 2004; Thonon et al., 2016), we chose two negative stereotypes regarding mental illness. We asked respondents to what extent they would agree or disagree with the following statements: “A person with (disorder in the vignette) is unpredictable” and “A person with (disorder in the vignette) is dangerous” Again, answers could be given on a 4-point Likert-scale (plus “don’t know” category).

Emotional reactions

Emotional reactions were assessed by eight items representing different ways of emotionally responding to the person in the vignette. Each item was coded from 1 “completely disagree” to 4 “completely agree.” Principal component analysis yielded the same three factors also found in previous research: anger, fear and prosocial reactions (Angermeyer, Holzinger & Matschinger, 2010). The items “I react angrily,” “I feel annoyed” and “This triggers incomprehension with me” loaded on the factor anger. The factor fear was comprised of the items “This triggers fear,” “I feel uncomfortable” and “I feel insecure,” while the items “I feel pity,” “I feel sympathy,” “I want to help” loaded on the factor prosocial. Together, the three factors accounted for a cumulative variance of 60.3% for schizophrenia and 59.1% for depression.

Desire for social distance

We assessed the respondents’ desire for social distance by means of a scale developed by Link et al. (1987), a modified version of the Bogardus Social Distance Scale (Bogardus, 1925). The scale includes seven items representing different social relationships, e.g. tenant, co-worker or child carer. On a Likert-scale ranging from 1 “completely disagree” to 4 “completely agree,” the respondents were asked to indicate to what extent they would accept the person with schizophrenia or depression described in the vignette. Non-linear principal component analysis was carried out, all items loaded on one factor. As we have reversed the scale, higher scores indicate greater desire for social distance.

Statistical analyses

Descriptive characteristics and bivariate associations were computed using SPSS 22 (SPSS Statistics for Windows. Version 22.0; IBM, Armonk, NY, USA) (For bivariate associations, please see tables in the additional material). We assessed the associations between stereotypes, continuum belief, emotional reactions and desire for social distance via path models computed in AMOS 22 (Arbuckle, 2013). Initially, an analysis of missing values was performed and data was imputed by means of full information maximum likelihood (FIML) in AMOS (Arbuckle, 1996; Enders & Bandalos, 2001). There are measures of “approximate model fit” that have been developed to avoid rejection of appropriate model structures due to large sample size (N > 300), e.g. the “root mean squared error of approximation” (RMSEA). Values of ≤ 0.05 indicate good model fit (Browne & Cudeck, 1993; Steiger, 1990). Additionally, the “Tucker Lewis index” (TLI) and the “Comparative fit index” (CFI) serve as indicators for good (≥ 0.95) model fit (Hu & Bentler, 1999). We started out from a saturated model and performed backward-selection of non-significant paths. The backward-selection was also applied to error terms to indicate whether we can assume shared unexplained variance. The significance level is set at α < 0.05; path analyses are adjusted for gender, age, level of education and identification of disease.

Results

The descriptive statistics for the scales and items indicating the stigma process are summarized in Table 2. Compared to depression, respondents were rather reluctant to agree with a continuity of symptoms in the case of schizophrenia. Regarding emotional reactions to a person with mental illness, prosocial feelings were most pronounced, followed by feelings of fear and anger, while the stereotype of unpredictability was endorsed more strongly than dangerousness in both cases. Regarding the results of the bivariate associations between continuum belief and stigma components for both disorders, please see the tables provided as additional material.

Table 2 Distribution of social distance, emotional reactions, stereotypes and continuum belief (mean value and standard deviation).

Items	Subsample schizophrenia (n = 1,338)	Subsample depression (n = 1,316)	
Desire for Social Distance1	19.84 (4.61)	15.78 (4.10)	
Emotional Reaction2	
 Anger	4.90 (1.74)	4.73 (1.74)	
 Fear	6.69 (2.27)	5.13 (1.92)	
 Prosocial	8.58 (1.69)	8.96 (1.71)	
Stereotype3	
 A person with (disorder in the vignette) is dangerous	2.47 (0.82)	1.83 (0.82)	
 A person with (disorder in the vignette) is unpredictable	3.07 (0.76)	2.37 (0.90)	
Continuum Belief3	2.13 (0.90)	2.89 (0.86)	
Notes:

1 Desire for social distance scale comprised of 7 items, ranging from 7–28.

2 Emotional reaction scales each comprised of 3 items, ranging from 3–12.

3 Stereotypes and continuum belief ranging from 1 “completely disagree” to 4 “completely agree.”

Table 3 displays fit measures of the path models as well as thresholds for good model fit. The χ2-values representing goodness of fit for the models were insignificant, thus data information is sufficiently explained by the models. The goodness of fit indexes displayed values of > 0.95 (TLI and CFI) respectively < 0.05 (RMSEA), indicating good model fit.

Table 3 Model fit.

	χ2	df	p	χ2/df	TLI1	CFI2	RMSEA3	
Thresholds	
 For good fit				≤ 2.0	≥ 0.95	≥ 0.95	≤ 0.05	
Path model schizophrenia	8.30	6	0.217	1.38	0.990	0.998	0.017 (90% CI = 0.000−0.042)	
Path model depression	6.75	7	0.456	0.96	1.00	1.00	0.000 (90% CI = 0.000−0.000)	
Notes:

1 Tucker Lewis index.

2 Comparative Fit index.

3 Root mean squared error of approximation.

In Fig. 2, the path model for schizophrenia is displayed (statistically significant paths only). Contrary to what we have expected, there are no associations between the stereotypes and the continuum belief. However, there are significant paths leading from the ascription of “unpredictability” to the emotional responses anger and fear, as well as to desire for social distance. These associations are positive, i.e. the more the respondents agree with the item “unpredictable,” the more they tend to react angrily or fearfully, and the greater their desire for social distance. The stereotype of “dangerousness” displays similar associations with anger and desire for social distance. Additionally, there is a negative association with prosocial reactions, i.e. the more respondents agree with the ascription of “dangerousness” to a patient with schizophrenia, the less they exhibit prosocial emotions. As expected, positive associations exist between anger, fear, and desire for social distance, while prosocial reactions are negatively associated with social distance. Continuum belief is associated with prosocial reactions (positively) and desire for social distance (negatively). I.e. those respondents, who rather agree with a continuum of symptoms, display greater prosocial emotions and less desire for social distance toward a person with schizophrenia.

Figure 2 Schizophrenia: path model of the relationship between stereotypes, continuum belief, emotional reactions and desire for social distance (standardized coefficients; significant paths only (α < 0.05); RMSEA: 0.017 (90% CI 0.000; 0.042)).

The path model for depression is presented in Fig. 3 (statistically significant paths only). As in the schizophrenia-model, there are no significant paths between stereotypes and continuum belief. The ascriptions of being unpredictable or dangerous are positively associated with feelings of anger and fear. In contrast to the schizophrenia-model, there are no significant associations between stereotypes and prosocial emotional reactions. Negative emotional reactions are associated positively with desire for social distance, while the association between prosocial feelings and desire for social distance is negative. As with the schizophrenia-model, belief in continuity of symptoms is positively associated with prosocial reactions and negatively with desire for social distance.

Figure 3 Depression: path model of the relationship between stereotypes, continuum belief, emotional reactions and desire for social distance (standardized coefficients; significant paths only (α < 0.05); RMSEA: 0.000 (90% CI 0.000; 0.033)).

Discussion

The aim of the present study was to explore associations of four consecutive components of the stigma process as postulated by Link & Phelan (2001) (Link et al., 2004) with a particular focus on belief in continuity of mental illness symptoms. The continuum belief served as a proxy for the step of separation between “us” and “them” in the stigma process. We focused our analyses on the stigma of persons with either schizophrenia or depression and explored associations between negative stereotypes (dangerous, unpredictable), continuum belief, emotional reactions (anger, fear, prosocial) and desire for social distance using a path modeling approach.

We expected continuum beliefs to be associated with the ascription of stereotypes as well as with emotional reactions and desire for social distance (see Fig. 1). However, there were no statistically significant relations between stereotypes ascribed to a person with mental illness and the belief in the continuity of symptoms. Moreover, the models revealed only two statistically significant associations regarding the continuum belief. The item was positively associated with prosocial emotional reactions and displayed a negative relation with desire for social distance. These results are in line with our expectations as well as with findings of other studies, which also showed that continuum belief was associated with more prosocial emotional reactions and less desire for social distance (Angermeyer et al., 2014b; Wiesjahn et al., 2014). Contrary to what we have expected, there were no statistically significant associations between feelings of fear or anger with continuum beliefs. As far as the other hypotheses are concerned, most expectations have been met. Both stereotypes ascribed to persons with mental illness were positively associated with desire for social distance and displayed relations with emotional reactions as described in step 5 in Fig. 1. These results are also supported by previous research. Stereotypes have shown to have a powerful impact on the preference to socially distance oneself from a person with schizophrenia (Angermeyer & Matschinger, 2004), and the associations between stereotypes and emotional reactions are also well researched (e.g. Angermeyer & Matschinger, 2003; Corrigan et al., 2002; Angermeyer & Dietrich, 2006). Similarly, the positive associations between feelings of anger and fear with social distance as well as a negative relation between positive emotions and preference to distance oneself have been discussed elsewhere (e.g. von dem Knesebeck et al., 2014). Regardless of the disease under study, we obtained similar results, which is remarkable and underlines the robustness of the findings.

In addition to the significant paths shown in Figs. 2 and 3, we would like to mention that there was an insignificant, but positive, relation between continuum beliefs and feelings of anger, which did not meet our expectations (see step 5 Fig. 1). This result is in line with other studies (Angermeyer et al., 2014b; Wiesjahn et al., 2014) and had been discussed by the authors as a negative consequence of losing the sick role as established by Parsons (1951). If mental health problems are rather seen on a continuum than being a dichotomy of health and illness, illness-related behavior of a person with mental illness could be less accepted. However, regarding the statistical insignificance of our results, we can only speculate on these associations between anger and continuum belief. The paradoxical contrariety of psychiatric stigma has also been considered by Gergel (2014). On the one hand, the stigma of mental illness is characterized by the idea of “otherness” caused by some biological flaw. A person with mental illness is seen as intrinsically different from oneself, which can facilitate negative feelings such as fear. This is referred to as unlikeness-based stigma by Gergel (2014). On the other hand, there is a second point of view on those with mental illness, which is paradoxically opposed to this “otherness,” termed likeness-based stigma. Assuming that a person with mental illness is not inherently different from oneself, sharing similar biological and environmental factors, can reinforce notions such as blame or anger, as it implies that a problem or deviant behavior lies in the responsibility of the person affected and is somehow a “weakness of character.” At the same time, this likeness or similarity also bears the potential to elicit feelings of fear or denial. Perceiving a person with mental illness as similar to oneself could be met with rejection as it implicates own vulnerability. Finding a solution to this paradox still represents a challenge in stigma research. A lot of research and anti-stigma work has attended to the unlikeness stigma, while the likeness-based stigma is also quite prevalent. Pescosolido & Martin (2015) emphasize that it is important to consider different, interrelated structures when it comes to understanding the complexity of stigma and its effects. In addition to that, it is of importance to bear in mind the two sides of mental illness stigma when it comes to the development of anti-stigma measures. Perhaps the combination of different approaches can be the right means to integrate both perspectives. On the one hand, conveying knowledge about the disorder. On the other hand, it has been found very important to facilitate personal contact to someone affected by mental illness (Griffiths et al., 2014).

When evaluating our findings, some limitations need to be mentioned and discussed. The continuum belief question was posed directly following the vignette, before respondents were asked about the disorder in question. If the interviewees could not identify the disorder correctly, they were given the answer by the interviewer. Thus, continuum notions evoked among respondents refer to the description of symptoms in the vignette. On the one hand, this is to prefer over diagnostically labeled vignettes as these could have reinforced the notion of differentness between the respondent and the person in the vignette. This might decrease the belief in the continuity of symptoms. On the other hand, correct identification of the mental health problem described in the vignette constitutes a possible confounder. This is why we have decided to additionally control our analyses for correct identification of the disorder. It has also to be kept in mind that beliefs elicited by a case vignette can be very distinct from attitudes displayed when actually meeting a person with mental illness. Furthermore, we also have to mention that there is a wide range of symptom presentations. Regarding the vignettes in our study, we only included core diagnostic signs and symptoms to keep the case story at a reasonable length. A central aspect that needs to be discussed is the absence of any statistically significant associations between stereotypes and continuum belief. This led us to the question whether using continuum belief as a proxy for the step of separation was the right approach. However, the continuum belief fits in the conceptualization of stigma as it stresses the similarity between persons with mental illness and those without, thus challenging a dichotomous view on health and illness or “us” and “them.” In alleviating perceived differentness between the in-group and the stigmatized group, the continuum belief-item seemed the most appropriate means. Moreover, the fact that we obtained similar results concerning the continuum belief, regardless of the disorder under study, corroborates the robustness of the model. Nevertheless, we also have to consider the fact that we only used a single item to assess agreement with the notion of continuity of symptoms. This may impede a comprehensive assessment of (dis-)agreement of continuity of mental illness symptoms in the public. In comparison to Wiesjahn et al. (2014), we only covered one possible dimension of the continuum belief. The single-item measure refers to the experience of a similar situation among the “normal” population, while the questionnaire used by Wiesjahn et al. (2014) covered three different dimensions: prevalence of symptoms in the “normal” population, clear distinction of categories vs. continuum, as well as a dimensional approach to mental illness, symptoms and associated distress. In addition, our results are limited to two negative stereotypes (unpredictability and dangerousness). A further methodological aspect that we have to consider is the response rate. A rate of 53% is satisfactory for telephone surveys in Germany (Schlinzig & Schneiderat, 2009), however, we cannot rule out a selection bias due to non-response. At the same time, a comparison with official statistics (Mnich et al., 2015) sustains the external validity of our study. It has to be kept in mind though, that our data is confined to the metropolitan regions of Hamburg and Munich and of cross-sectional nature. We are aware that there can be possible differences in composition of social levels and/or population age when comparing metropolitan to rather rural areas in Germany. However, due to the campaign’s design a certain focus on Hamburg and its adjacent areas was necessary.

To our knowledge, this is the first study that investigated continuum beliefs in the context of the stigma process approach as postulated by Link & Phelan (2001) (Link et al., 2004). The associations found support a possible destigmatizing effect of the belief in continuity of symptoms. Promulgating continuum belief as a means to oppose perceived separation between “us” and “them” could prove useful for future anti-stigma campaigns. This is corroborated by an experimental study by Wiesjahn et al. (2016), in which a continuum belief intervention was consistently associated with lower stereotypes, less fear and decreased desire for social distance. Schomerus et al. (2016) came to similar results. In an online survey experiment, the conveyance of information on a mental health-mental illness continuum led to decreased perceived difference and increased social acceptance. Integrating a continuum approach into existing measures to encounter psychiatric stigma would be feasible, e.g. when it comes to work in small to medium sized groups where dissemination of knowledge is combined with personal contact to someone with mental illness, a form of intervention that has been shown to have great effects on stigmatizing attitudes (Griffiths et al., 2014).

Supplemental Information

Supplemental Information 1 Raw data: schizophrenia.

Click here for additional data file.

Supplemental Information 2 Raw data: depression.

Click here for additional data file.

Supplemental Information 3 Annex Correlation Analyses.

Click here for additional data file.

Supplemental Information 4 Correlation analyses: schizophrenia.

Click here for additional data file.

Supplemental Information 5 Correlation analyses: depression.

Click here for additional data file.

Psychenet is a project network in the region of Hamburg which consists of more than 80 scientific and medical institutions, counselling centers, the Senate and the Chamber of Commerce of the Free and Hanseatic City of Hamburg, companies, as well as patients’ and relatives’ associations (2011–2015). The vision of the project is to promote mental health today and in the future, concerning early diagnosis and effective treatment of mental illnesses. Coordination of the joint project is mutually carried out by the Gesundheitswirtschaft Hamburg GmbH and the University Medical Center Hamburg-Eppendorf. For more information and a list of all partners, please visit www.psychenet.de.

Appendix

Vignettes

Schizophrenia*

For around half a year now, 19-year-old Sabine S. retreats more and more in herself, avoids all contacts and has the impression that other people can read her mind. She is often scatterbrained and absent-minded. For some time now, Sabine has been feeling threatened and persecuted. She also hears voices which disrupt her thoughts and give her instructions.

Depression*

46-year-old Dagmar D. has been constantly downhearted and unhappy for the last few months. She worries about the future. Mrs. D. feels useless, has the impression everything she does is wrong and has lost all interest in everyday activities. Besides, she complains about insomnia and feels nerveless and weak, already in the mornings. Mrs. D’s capability to work turns out to be declining.

*Gender of the “patient” was systematically varied.

Additional Information and Declarations

Competing Interests

Author Contributions

Ethics

Data Deposition

The authors declare that they have no competing interests.

Anna C. Makowski conceived and designed the experiments, analyzed the data, wrote the paper, prepared figures and/or tables.

Eva E. Mnich conceived and designed the experiments, prepared figures and/or tables, reviewed drafts of the paper.

Matthias C. Angermeyer contributed reagents/materials/analysis tools, reviewed drafts of the paper.

Olaf von dem Knesebeck conceived and designed the experiments, analyzed the data, wrote the paper, prepared figures and/or tables, reviewed drafts of the paper.

The following information was supplied relating to ethical approvals (i.e., approving body and any reference numbers):

The study was approved by the Ethics Commission of the Medical Association in Hamburg, Germany (PV3707).

The following information was supplied regarding data availability:

The raw data has been supplied as Supplemental Dataset Files.

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
