# Peer review of "Continuum beliefs in the stigma process regarding persons with schizophrenia and depression: results of path analyses"

_PeerJ, doi:10.7717/peerj.2360_

## Round 0.1 · original submission · Minor Revisions

In my capacity as an Academic Editor for your article, a role I agreed to take on when a colleague on the Board closer to your field of research was unable to accept the role, I have read your article and the comments of the referees. While outside the field, I am nonetheless a Trustee of a major Mental Health Research Charity in the UK and so familiar with discussions about stigma.

My understanding is that, with references to the continuity of mental illness symptoms (serving as a proxy for separation), you sought to explore its associations with stereotypes, emotional responses and desire for social distance in the stigma process. You discuss an extensive questionnaire conducted by telephone and present the data you secured using this approach in relevant detail. Certain predictions you made at the outset were not up-held but you nonetheless secured interesting findings. Specifically, you have grounds for suspecting that including information on the continuity of symptoms could prove helpful in future anti-stigma campaigns.

Both referees, who are well within your field of expertise, recommend only minor revisions, with one explicit about recommending publication. They may perhaps be suggesting more revision than is really necessary but, as this is outside my expertise, I have decided to allow you to choose amongst their suggestions those which you judge are most relevant to improving the manuscript.

I have no comments on the Statistics and believe your use of SPSS is entirely appropriate.

Finally, we apologise the length of time getting back to you has taken, but we have gone to considerable lengths to secure what we hope are appropriate referees. We hope that, in the circumstances, you will be tolerant of the delay.

RM

Reviewer 1 ·

Basic reporting

The manuscript “Continuum beliefs in the stigma process regarding persons with schizophrenia and depression: results of path analyses” investigated continuum beliefs about depression and schizophrenia in the context of the stigma model postulated by Link and Phelan (2001). Continuum beliefs serve as a stand in for the stigma component of cognitive separation between “us” and “them”. The hypotheses regarding continuum beliefs are confirmed for their association with discrimination (i.e. desire for social distance) and pro-social attitudes, however no associations with stereotypes and negative emotional reactions were found.

The article is well written, the background and importance of the study are clearly presented in the introduction. The structure conforms to PeerJ standard and figures are relevant to the article and well labeled and described.

Experimental design

The submission clearly defines a relevant and meaningful research question. Methods are described in sufficient detail and adherence to ethical standards is documented.

Validity of the findings

With respect to the results and their interpretation, I have some minor questions and concerns.

a. As shown in Figures 2 and 3, error terms for “Anger”, “Fear”, and “Prosocial Reaction” are allowed to correlate, while there appears to be a simple association of the indicators “Unpredictable” and “Dangerous”, the reason for this is not explained. It seems counterintuitive that error term correlation would be included in a “backward-selection” (line 142) by eliminating non-significant paths. A brief explanation for this may be warranted.

b. lines 209-226: The authors elaborate on the association of continuum beliefs and anger towards people diagnosed with a mental disorder. Given that the corresponding paths in both models (Figure 2 and 3) are non-significant in what are large samples, these elaborations seem not be supported by the data. Furthermore, the absolute values of these positive associations are not reported. I sympathize with the authors’ intention: other studies found an association between anger and continuum beliefs and it seems prudent not to ignore potential negative side effects something that is already tested as a potential anti-stigma message. However, given that the data shows no such negative associations, I suggest the respective section to be formulated more clearly as a speculation rather than something that is drawn from the study at hand.

c. I found the discussion of the limitations quite interesting (lines 249-252). The authors speculate that the one-item measure may have been a methodological limitation. If I understand correctly, they point towards the curious finding that associations with stereotypes have been found most consistently when questionnaires have been used to assess continuum beliefs, as they hinted at in the introduction (lines 54-57). I wonder if the authors may have an idea, why this is the case - or more generally: What they expect to be potentially occluded due to a one item assessment of continuum beliefs.

Additional comments

In sum, the manuscript addresses the timely matter of utilizing the continuum model in stigma research and provides an empirical test of a comprehensive model with a large sample size and for different categories of mental illness. Given the authors address my aforementioned concerns, I am happy to recommend it for publication.

Reviewer 2 ·

Basic reporting

The present study deals with the important topic of stigmatization against persons with mental disorders and adds to the small number of studies dealing with continuum beliefs. The manuscript is well written and the literature is discussed well.

The figures illustrate the hypotheses and results; however, it would be helpful to include the expected associations (+ or -) attached the arrows in figure 1 (and leave out lines 66 to 74).

Furthermore it would be helpful to include an additional table with the correlations between the variables in order to compare the results to previous studies.

In table 1 the authors could add the means and standard deviations for age and years of education.

In table 2: The authors could add information on the statistical comparison between the subgroups.

Experimental design

The authors develop the research question clearly from previous literature. However, they could add some more information on previous anti-stigma strategies. Their lack of effect (or even detrimental effect) points to the need of improving anti-stigma approaches.

The design and the sample are suitable for the research question. The authors included a large and representative sample. But there are some limitations that need to be kept in mind:
- Small number of items for some variables (cb, stereotypes)
- Possible selection effect (metropolitan areas, response rate of 50%)
- Cross-sectional design based on correlations

The design is described briefly. The authors could add more information on the procedure (lines 90 ff): Did the vignettes include diagnostic labels? Order of the questions etc.

Validity of the findings

The authors present a good discussion regarding the “paradoxical contrariety” (l. 215 ff). Following this it would be great if they could add possible solutions for this dilemma.

In contrast to other literature (eg Wiesjahn 2014) they found no association between cb and stereotypes; they could add some more possible explanations on this.

They should discuss the possible selection effect (metropolitan areas, response rate) more closely. What would they expect in a more rural or heterogeneous sample?

Furthermore they should discuss the cross-sectional design: What are the limitations? What could be alternative “stronger” designs?

Additional comments

No comments.

---

## Round 0.2 · accepted · Accept

The revised manuscript is accepted for publication.